

# Screw dislocations in the X-cube fracton model

Nandagopal Manoj[1], Kevin Slagle[2,3], Wilbur Shirley[2] and Xie Chen[2,3]

**1** Indian Institute of Science, Bangalore 560012, India
**2** Department of Physics and Institute for Quantum Information and Matter,
California Institute of Technology, Pasadena, California 91125, USA
**3** Walter Burke Institute for Theoretical Physics,
California Institute of Technology, Pasadena, California 91125, USA

## Abstract

The X-cube model, a prototypical gapped fracton model, was shown in Ref. [1] to have a foliation structure. That is, inside the $3 + 1$D model, there are hidden layers of $2 + 1$D gapped topological states. A screw dislocation in a $3 + 1$D lattice can often reveal non-trivial features associated with a layered structure. In this paper, we study the X-cube model on lattices with screw dislocations. In particular, we find that a screw dislocation results in a finite change in the logarithm of the ground state degeneracy of the model. Part of the change can be traced back to the effect of screw dislocations in a simple stack of $2+1$D topological states, hence corroborating the foliation structure in the model. The other part of the change comes from the induced motion of fractons or sub-dimensional excitations along the dislocation, a feature absent in the stack of $2 + 1$D layers.

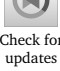

# 1   Introduction

Fracton models [2–8] are characterized by the peculiar feature that some of their gapped point excitations are completely localized or are restricted to move only in a lower dimensional sub-manifold. The X-cube model, first proposed in Ref. [5], is one of the most widely studied gapped fracton models in $3+1$D. It captures many important features of gapped type I fracton models, including a ground state degeneracy that grows exponentially with linear system size, the existence of fracton and other sub-dimensional fractional excitations, and subleading linear entanglement scaling [9–11]. In particular, it was shown in Ref. [1] that the X-cube model has a foliation structure [9,12–16]. That is, starting from the ground state of the X-cube model on a 3D cubic lattice with periodic boundary conditions, a 2D toric code state can be decoupled from the 3D bulk using a finite depth local unitary circuit near the two dimensional layer, such that the remaining 3D bulk is still the X-cube model but of one lattice spacing smaller. There are hence a large number of hidden layers of toric code inside the X-cube model, giving rise to several of the properties mentioned above: a linearly growing number of ground space logical qubits , the existence of planons (i.e. fractional excitations that move in planes), and sub-leading linear entanglement scaling. The layers in a foliation structure are called 'leaves'.

In systems with a layered structure, nontrivial features can often be revealed by inserting a screw dislocation through the layers. For example, a weak 3+1D topological insulator is equivalent to a stack (or several stacks) of 2+1D topological insulators [17]. It was shown in Ref. [18], that a screw dislocation in a weak topological insulator carries topologically protected 1D gapless fermionic excitations. As shown in Fig. 1, a screw dislocation through a stack of 2D layers connects all the layers together and the screw dislocation becomes one edge of the expanded annulus. If the 2D layers host gapless edge states (as in the case of topological insulators [19] and chiral topological states [20]) [21–24], the screw dislocation should carry the gapless mode along its length. If the gapped 2D layers can have gapped edges due to anyon condensation (as in the case of 2D toric code [25] and other non-chiral topological states [26, 27]), then the screw dislocation can be gapped as well, potentially leading to topological degeneracy if the condensation matches that at the outer boundary.

Given the foliation structure in the X-cube model made up of 2D topological layers, we can ask whether similar nontrivial features exist along screw dislocations. As the 2D layers in this case host nonchiral topological order, are there extra topological degeneracies associated with the screw dislocation? Indeed, this is what we find in this work. We see through direct lattice calculation that when a screw dislocation is inserted into the X-cube model on a regular cubic lattice, it can result in extra topological degeneracy. The results are summarized in Fig. 2. We find that when the boundary condition at the dislocation matches with that on the outer boundary, a screw dislocation can introduce extra ground state degeneracy (GSD) compared

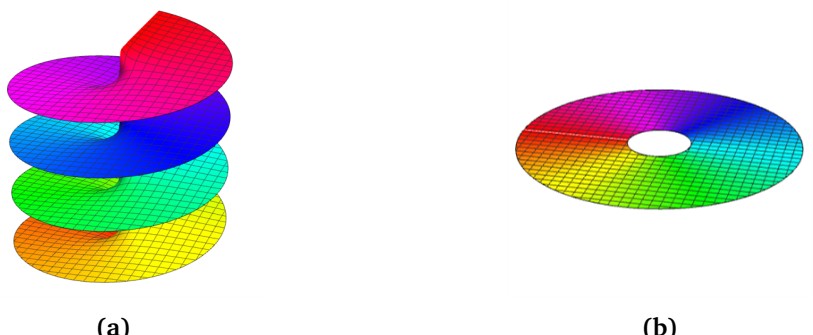

**(a)**                                **(b)**

Figure 1: **(a)** A screw dislocation through a stack of 2D layers connects the layers together. **(b)** The structure in (a) can be expanded into an annulus on a single plane, with the screw dislocation becoming the inner boundaries.

to a hole in the system. The change in ground state degeneracy comes from two sources (or logical operators): 1. the winding of planon quasi-particles (i.e. fractional excitations that move in planes) around the screw dislocation as it connects the foliation layers in the system (the +1 part); 2. the tunneling of fracton or lineon quasi-particles (i.e. fractional excitations that only move along lines) along the screw dislocation (the (+1) part). (Fractons, which are usually immobile, can gain some mobility near certain kinds of dislocations.) The latter effect has an even/odd dependence on the length of the defect line and reflects the nontrivial fractonic nature of the X-cube model beyond the foliation structure. When the boundary conditions do not match, the tunneling becomes trivial and there is no change in GSD associated with the screw dislocation.

To calculate the GSD, we can make use of the foliation structure in the model. In particular, we can keep decoupling 2D topological layers from the 3D bulk with unitary transformations until a *minimal structure* is reached, such that no more layers can be removed. The log (with base 2) of the total GSD is a sum of the log degeneracy in each layer and that of the minimal structure. For example, consider the X-cube model on 3D cubic lattice with periodic boundary

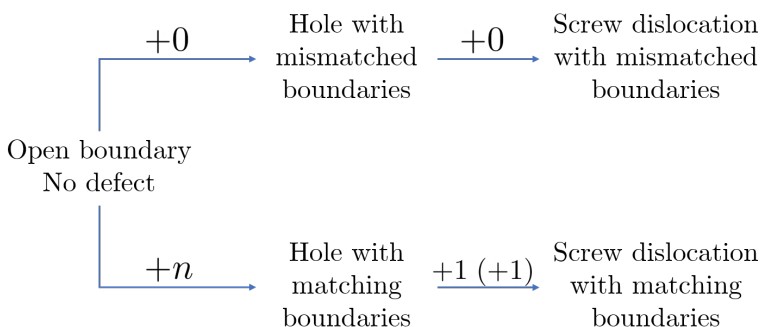

Figure 2: Change of $\log_2$ GSD due to screw dislocation. When boundary condition at the dislocation matches with that on the outer boundary, an edge dislocation/hole introduces extra GSD that increases with the size of the defect ($+n$). Moreover, a screw dislocation can introduce extra GSD due to the tunneling of planon quasi-particles around the screw dislocation as it connects the foliation layers in the system (the +1 part) and the tunneling of fracton or lineon quasi-particle along the screw dislocation: the (+1) part. The latter depends on the size of the defect line. When the boundary conditions do not match, there is no change in GSD: +0.

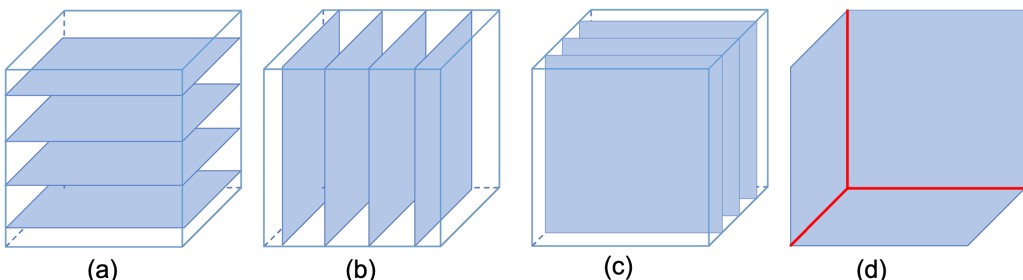

Figure 3: Procedure for calculating the GSD of the X-cube model on the 3-torus. (a-c) Decoupling 2D toric code layers (via a local unitary transformation) from the 3D bulk in $xy$, $yz$ and $zx$ planes. After removing all the foliation layers, the remaining minimal structure is composed of a 2D toric code layer in each direction, which are then coupled along the red intersection lines.

conditions. Each decoupled layer is a 2D toric code with periodic boundary conditions, which contributes a $\log_2$ GSD of 2. The decoupling procedure can be continued until we are left with one leaf each in the $xy$, $yz$ and $zx$ planes, respectively. Such a minimal structure can be thought of as three 2D toric code states, in $xy$, $yz$, $zx$ planes respectively, coupled along their intersection lines [28, 29] such that the Wilson loops on intersecting planes merge into one. Because of the coupling, the $\log_2$ GSD of the minimal structure is $3 \times 2 - 3$. Therefore, altogether, the $\log_2$ GSD of the cubic lattice model is equal to $2L_x + 2L_y + 2L_z - 3$. This procedure is graphically illustrated in Fig. 3 and detailed in the next section. We will apply this procedure to a variety of other lattice structures, including ones with open boundary conditions, with holes and with screw dislocations.

The paper is structured as follows: First, we review the X-cube model on a cubic lattice and discuss its general properties. We then proceed to describe the minimal structure approach to calculate the ground state degeneracy of the X-cube model with and without boundaries, discussing how the Wilson loops bind together in each case (Sec. 2). Then, we look at a 'smooth' screw dislocation, and derive its ground space properties explicitly using the minimal structure approach (Sec. 3). We discover that the dislocation provides increased mobility to the subdimensional excitations, and this results in an increased ground state degeneracy. To obtain a more complete picture of dislocation defects in the X-cube model, in Sec. 4, we take a look at other crystal defects such as holes and edge dislocations, and analyze them using the underlying foliation structure of the model. We argue that a general screw dislocation can be thought of as a hole plus a screw dislocation, which gives us a straightforward process to obtain the GSD of the X-cube model on a lattice with a general screw dislocation. Finally, we discuss what happens when the screw dislocation has larger Burgers vectors. We summarize our findings in Table 1.

## 2 Boundary conditions

In this section, we will review the basic properties of the X-cube model and discuss how to calculate the ground state degeneracy in the cases of periodic and open boundary condition (no dislocations) using the procedure described above.

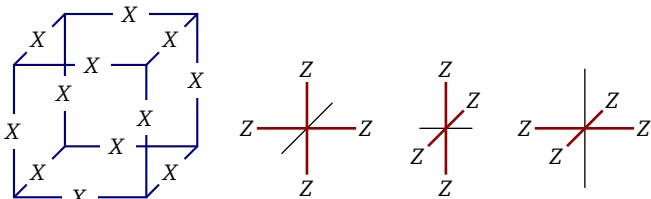

Figure 4: Cube ($A_c$) and vertex ($B_v^\mu$) operators of the X-cube model Hamiltonian on a cubic lattice.

## 2.1 Review: the X-cube model

The X-cube model [5] is a gapped fracton model with spin-$\frac{1}{2}$ degrees of freedom (qubits) living on the links of a simple cubic lattice. It has a Hamiltonian of the form

$$H = -\sum_c A_c - \sum_{v,\mu} B_v^\mu, \tag{1}$$

where $A_c$ is the product of $\sigma^x$ (henceforth $X$) over all links on the cube $c$ and $B_v^\mu$ is the product of $\sigma^z$ (henceforth $Z$) over the links around the vertex $v$ such that the links are perpendicular to the direction $\mu = x, y, z$ (Fig. 4). Since $X$ and $Z$ anticommute, and $B_v^\mu$ and $A_c$ always have an even number of links in common, all terms in the Hamiltonian commute with one another. Therefore, this Hamiltonian forms a *stabilizer code* [30] and the ground space will be the simultaneous positive eigenspace of all these individual operators. The ground state degeneracy of the X-cube Hamiltonian is given by

$$\log_2 \text{GSD} = 2L_x + 2L_y + 2L_z - 3, \tag{2}$$

for the system on a 3-torus of size $L_x \times L_y \times L_z$.

This model has the interesting property that it has gapped excitations with restricted mobility. An isolated cube excitation (violation of the $A_c$ term) is a 'fracton' (a fractional excitation that cannot move) while an isolated vertex excitation is a 'lineon' (a fractional excitation confined to move in a one-dimensional submanifold). The cube excitations can be created and

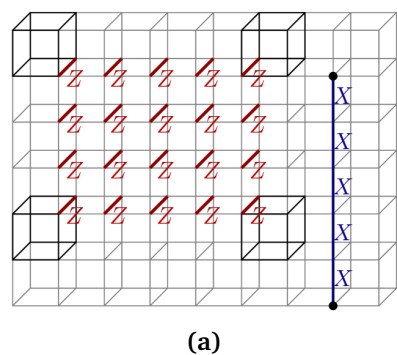

(a)

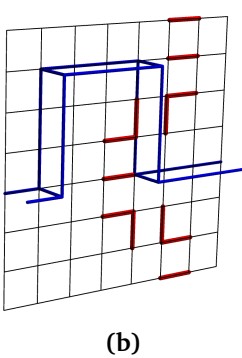

(b)

Figure 5: **(a)** Visualization of particle creation operators in the X-cube model. The red links correspond to a membrane geometry on the dual lattice. The product of $Z$ operators over these edges excites the (darkened) cube operators at the corners. The product of $X$ operators over the links comprising the straight open blue string creates excitations at its endpoints (black dots); **(b)** Motion of dipoles of vertex and cube excitations in a plane is achieved by applying $X$ on the blue links and $Z$ on the red links respectively.

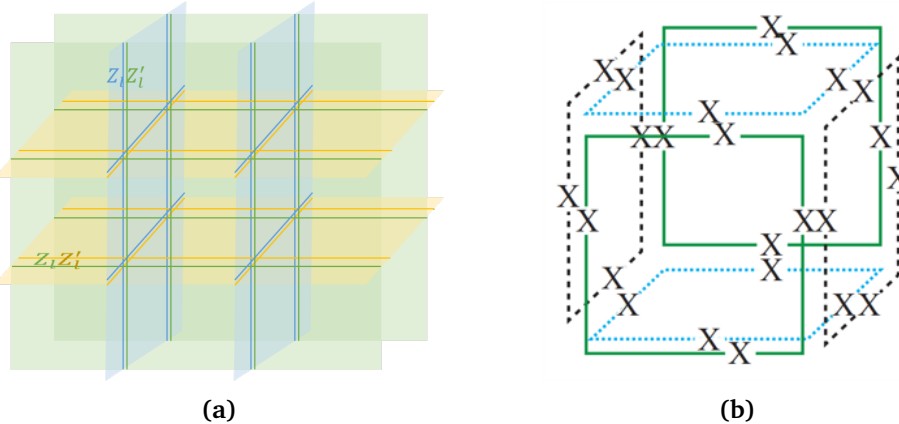

Figure 6: **(a)** Coupling stacks of toric code layers via a $ZZ$ interaction between overlapping links; **(b)** The form of $A_c$ which comes at sixth order in perturbation theory.

separated by applying $Z$ operators over a membrane on the dual lattice, and the vertex excitations can be created by applying $X$ operators on a line as shown in Fig. 5a. Dipoles of these fractons and lineons are mobile within a plane when isolated, which earns them the name planons. We will use the term fracton for the cube excitation, lineon for the vertex excitation, and planon for both fracton and lineon dipoles.

Another aspect of the X-cube model that is relevant to our discussion is the coupled layer construction of the model. It has been shown that the X-cube model can be obtained starting from three perpendicular stacks of toric code layers with a strong $ZZ$ coupling between overlapping links (Fig. 6a), dubbed the coupled layer construction [28, 29]. This is given by the Hamiltonian

$$H = \sum_{\mu=x,y,z;i} H_{\text{TC}(\mu,i)} - J \sum_l Z_l Z_l', \tag{3}$$

where $H_{\text{TC}(\mu,i)}$ refers to the Hamiltonian of the $i$th toric code stacked perpendicular to $\mu$. In the limit $J \gg 1$, this reduces to the commuting Hamiltonian (at sixth order in perturbation theory, refer Fig. 6b)

$$H = -J \sum_l Z_l Z_l' - \sum_{\nu,\mu} B_\nu^\mu - \mathcal{O}\left(J^{-5}\right) \sum_c A_c. \tag{4}$$

Here, $B_\nu^\mu$ is the vertex term of the toric code perpendicular to $\mu$ at the vertex $\nu$ in the simple cubic lattice, and $A_c$ is the product of $X$ on all qubits on the edges surrounding the cube $c$, as shown in Fig. 6b. We recover the X-cube model by looking at the $Z_l Z_l' = 1$ subspace of the full Hilbert space.

Each toric code has two logical qubits in its ground space, acted on by logical operators which correspond to operators that take vertex and plaquette excitations around the nontrivial cycles of the torus, which we will call Wilson and 't Hooft loops respectively, in analogy with $\mathbb{Z}_2$ gauge theory. This coupling binds together the Wilson loops of the intersecting toric code layers, because the individual Wilson loops do not commute with the $Z_l Z_l'$ term. Combining this feature with the foliation structure of the model gives us a way to derive the GSD of the model on various lattices, which we will show in detail in the following sections.

## 2.2 Periodic boundary conditions

In this section, we illustrate the minimal structure method to calculate the ground state degeneracy of the X-cube model on the 3-torus. This method will be applied to lattice structures

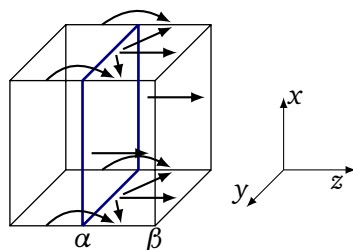

Figure 7: The finite depth local unitary to remove an $xy$ toric code leaf (the $\alpha$ plane highlighted in blue) from an X-cube model. The arrows denote CNOT gates going from the control qubit to the target qubit. We should apply this process everywhere on the $\alpha$ plane.

with boundaries and dislocation defects in later sections. The GSD of the X-cube model on a lattice with a given foliation structure is calculated by exfoliating leaves until we reach the minimal structure of the foliation. The minimal structure is significantly simpler than the original model and usually has a nice interpretation as a small number of toric code leaves coupled together. Since unitary circuits preserve the eigenvalues, we can combine the GSD of the exfoliated leaves and the minimal structure to find the GSD of the X-cube model. It is important to note that to reach this minimal structure we generally need to apply local unitary circuits with depth that scales as linear system size. So the minimal structure and decoupled toric codes are not in the same phase and do not have the same long range entanglement as the model we started with.

Let us consider the X-cube model on a 3-torus. As shown in Ref. [1], a local unitary circuit as defined in Fig. 7 can be used to remove toric code leaves from a large X-cube model until

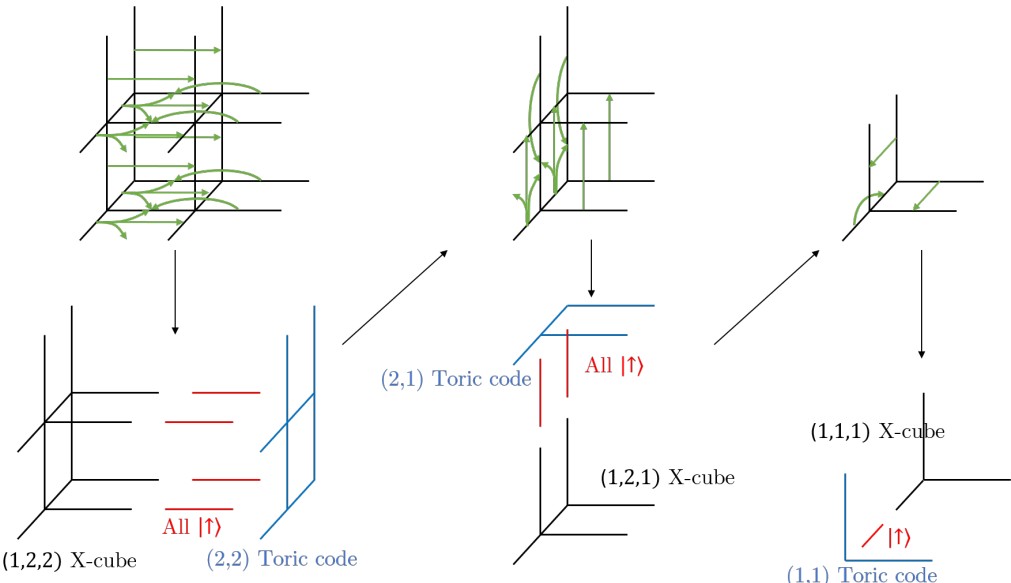

Figure 8: The unitary transformation to reach the minimal structure of the X-cube model on $T^3$. The arrows denote CNOT gates going from the control qubit to the target qubit, and one can verify that all the CNOT gates in each picture commute. Note that all the X-cube models and exfoliated toric codes at each step have periodic boundaries, so some of the CNOT arrows which seemingly do not point to any link actually point to the link on the other side of the torus.

we reach a system size of $2 \times 2 \times 2$. After this, we have shown the process explicitly to reach the $1 \times 1 \times 1$ minimal structure of the X-cube model on a 3-torus in Fig. 8. We observe that the $1 \times 1 \times 1$ X-cube model has no nontrivial stabilizer terms and hence has $\log_2 \text{GSD} = 3$. Using the fact that each exfoliated toric code has a GSD of 4, we can conclude that

$$\log_2 \text{GSD} = 2L_x + 2L_y + 2L_z - 3, \tag{5}$$

for an X-cube model on a $L_x \times L_y \times L_z$ cubic lattice with periodic boundaries.

The GSD of the minimal structure can be interpreted using the coupled layer picture. We see that, at each intermediate step of the exfoliation procedure, the model retains a coupled layer structure, but with fewer leaves. At the minimal structure with system size $1 \times 1 \times 1$, we can interpret the model as composed of three perpendicular toric code leaves coupled along their intersection lines. These toric codes have two independent Wilson loops each, but now they do not commute with the $ZZ$ coupling term along the three axes in the Hamiltonian. To make them commute, the two Wilson loops along each intersection line bind together, which reduces the number of independent Wilson loops to three, hence the $-3$ in the $\log_2 \text{GSD}$.[1]

## 2.3 Open boundary conditions

Now let us consider the X-cube model on a cubic lattice with open boundary conditions.

### 2.3.1 Smooth and rough boundary conditions

In this work, we consider the two simplest kinds of open boundary for the X-cube model, which can be characterised by the type of quasi-particles condensed at the boundary [33]. Analogous to the 2D toric code [25], these boundaries are called smooth and rough, where a smooth boundary condenses fracton dipoles and a rough boundary condenses the lineon and hence lineon dipoles. The fracton dipole corresponds to the charge excitation in the toric code foliation leaves while the lineon dipole corresponds to the flux excitation in the leaves. They are both planons and the smooth and rough boundary conditions have a straight forward correspondence with the smooth and rough boundary conditions in the toric code leaves.[2]

---

[1]It is also possible to understand the minimal structure in terms of the defect network construction of the X-cube model. [31, 32]

[2]Other boundary conditions of the X-cube model exist where different composite excitations are condensed at the boundary. Ref. [33] studies four kinds of gapped X-cube boundaries.

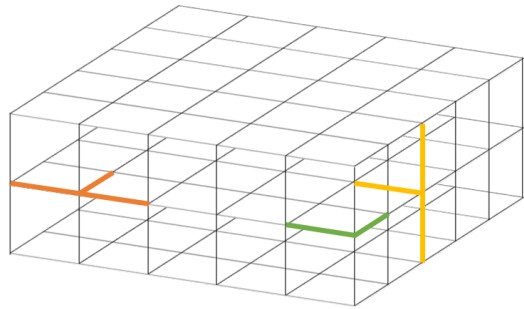

**(a)** Smooth boundaries. Product of $Z$ over the different sets of colored links give the corresponding vertex terms on the boundary.

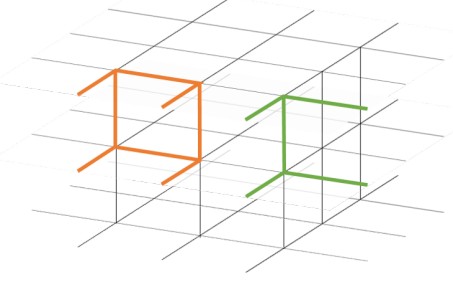

**(b)** Rough boundary along two axes and smooth along the third. The product of $X$ over the sets of colored links give the respective cube terms on the boundary.

Figure 9: Geometry of smooth and rough boundaries in the X-cube model.

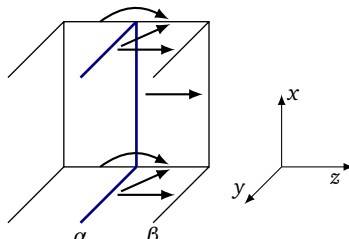

Figure 10: The finite depth local unitary to remove an $xy$ toric code leaf ($\alpha$) from an X-cube model near a rough boundary along the $xz$ plane.

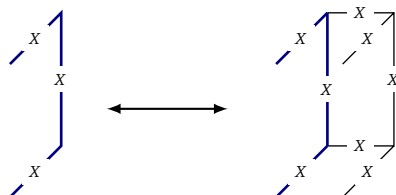

Figure 11: Adjoint action of the circuit on stabilizers of the two systems converts the plaquette term at the rough boundary of the toric code to the cube term in the X-cube model and vice versa. As in Fig. 10, bold (blue) lines correspond to edges of the new leaf.

Near a smooth boundary in the $xz$ or $yz$ plane, the vertex terms $B_v^{xz}$ and $B_v^{yz}$ only have three links in them (as shown in Fig. 9). Analogously, near a rough boundary, the cube terms only have eight links (instead of 12). The number of links in the term decreases even further when we reach an edge (corner) connecting two (three) perpendicular boundaries. It can be checked that a fracton dipole can disappear if it is brought close to a smooth boundary while a lineon or lineon dipole can disappear if it is brought close to a rough boundary.

### 2.3.2 Foliation structure

We will focus on systems with periodic boundaries along one direction (which we will choose to be $z$), and open boundaries along $x$ and $y$. From now on, unless specified otherwise, an X-cube model with smooth boundaries refers to a system with smooth boundaries along $x$ and $y$ and periodic boundaries along $z$, and similarly for rough boundaries. We will show that this has a similar foliation structure as discussed earlier, but the toric code leaves now have open boundaries. Because of this, the ground state degeneracy (GSD) of an X-cube model with boundaries will be very different from an X-cube model on a 3-torus. Since our system has similar open boundaries (smooth or rough) along $x$ and $y$, the GSD of the exfoliated 2D toric codes will be given by

$$\log_2 \text{GSD} = \begin{cases} 1 & yz \text{ plane} \\ 1 & zx \text{ plane} \\ 0 & xy \text{ plane} \end{cases}. \tag{6}$$

The first two cases correspond to a 2D toric code with periodic boundary condition along one direction and open boundaries along the other, with the same boundary type (smooth/rough) on both ends. This system has a two-fold ground state degeneracy. The last leaf is a toric code on an open disc with the same boundary all-around which is non-degenerate. Because of the foliation structure, we see that for an X-cube model on this geometry

$$\log_2 \text{GSD} = L_x + L_y + \text{O}(1). \tag{7}$$

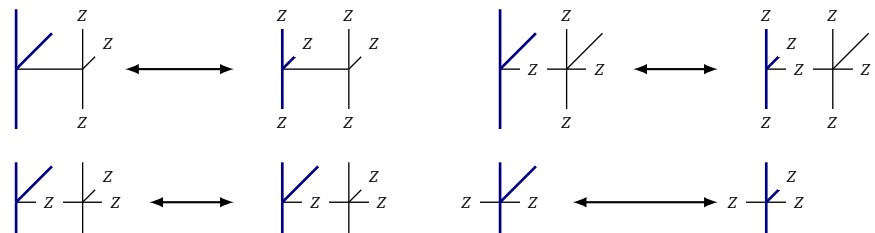

Figure 12: Adjoint action of the circuit on the stabilizers of the two systems in the case of a smooth boundary. Bold (blue) lines correspond to edges of the new leaf. The plaquette/cube terms transform exactly like the case without boundaries.

We can *exfoliate* toric code leaves from these systems using a finite depth local unitary similar to Fig. 7. The process is almost identical and differs from the standard process only near a rough boundary, which is shown in Fig. 10. The transformation of the stabilizer terms near the rough (smooth) boundary under the action of this unitary is shown in Fig. 11 (Fig. 12).

### 2.3.3 Minimal structure: smooth boundaries

Just like for the 3-torus, one can imagine performing this exfoliation process until we reach the minimal remaining structure. In Sec. 2.3.2, we discussed how to remove a leaf of toric code from an X-cube model with boundaries. We can apply that process to reduce an $L_x \times L_y \times L_z$ X-cube model to a $2 \times 2 \times 2$ X-cube model, along with $L_x - 2$ $yz$-plane (each with GSD 2), $L_y - 2$ $xz$-plane (each with GSD 2), and $L_z - 2$ $xy$-plane (with no GSD) toric code leaves. For smooth outer boundaries, the finite depth unitary transformation in Fig. 13 further decouples this into three additional tiny toric code leaves (with total GSD 4) and a $1 \times 1 \times 1$ X-cube model. Thus, the minimal structure is a $1 \times 1 \times 1$ X-cube model, which has just one qubit and no nontrivial stabilizer elements (i.e. its Hamiltonian is $H = 0$). Therefore, the GSD of the minimal system is 2 and it follows that for the original $L_x \times L_y \times L_z$ X-cube model on these boundary conditions:

$$\log_2 \text{GSD} = L_x + L_y - 1. \tag{8}$$

We can interpret the GSD by looking at the coupled layer construction of the minimal structure. The minimal structure can be interpreted as three transversely intersecting toric codes (with appropriate boundaries) strongly coupled together via a $ZZ$ type interaction for overlapping links. In the case of smooth boundaries, we see that the $xy$ leaf has no logical operator while the other two leaves consist of a single (pair of) logical operator, represented by the Wilson loop that winds the $e$ particle around the periodic boundaries. But, because of this strong coupling, these Wilson loops individually are not logical operators anymore, only the product of the Wilson lines from the two toric codes overlapped on each other is. Therefore there is only one logical qubit in the ground space. This is what gives rise to the '$-1$' in Eq. (8).

### 2.3.4 Minimal structure: rough boundaries

We can do a similar transformation to the X-cube model with rough boundaries to reach its minimal structure, which is the $1 \times 1 \times 1$ X-cube model with rough boundaries (Fig. 14). Note that we have periodic boundaries along $z$. This system has 5 qubits in the Hilbert space with a ground state degeneracy of $2^2$. This ground state degeneracy can be arrived at using the logical operators of the coupled toric code layers. In this case, it is easier to look at the 't Hooft loops of the underlying toric codes. We see that the $xz$ (blue) and the $yx$ (green) planes have a 't Hooft loop going around the periodic boundaries which forms a logical operator. This still commutes with the effective Hamiltonian Eq. (4) and they generate all the independent vertex

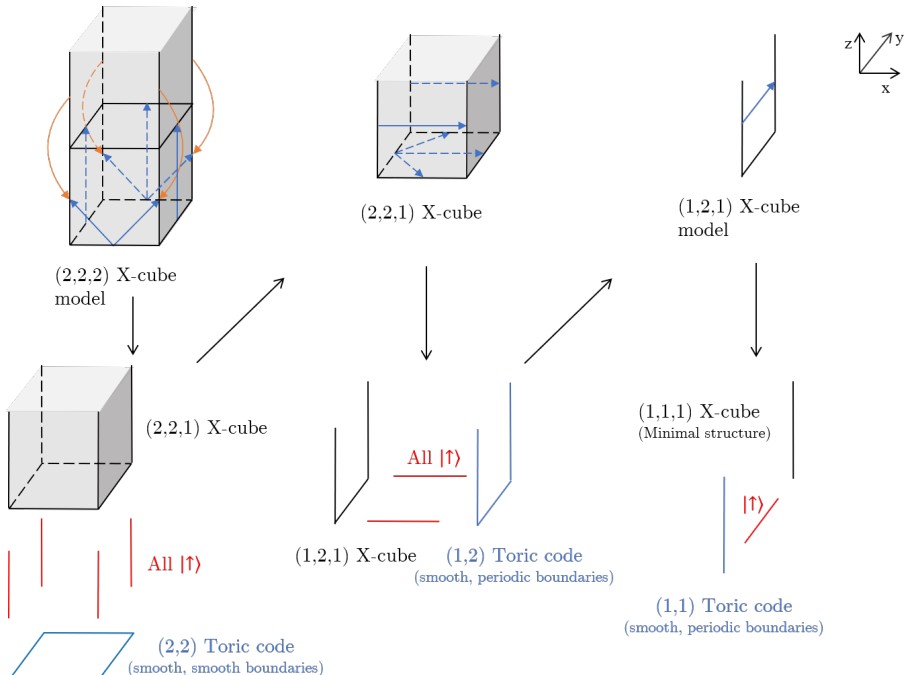

Figure 13: Using local unitaries to reach the minimal structure of the X-cube model with smooth boundaries along $x, y$ and periodic boundaries along $z$. The last local unitary (just the single CNOT gate) commutes with the $2 \times 1 \times 1$ X-cube Hamiltonian and doesn't change the ground space, but we are including it to show the straightforward pattern with which one can exfoliate (even small) leaves.

logical operators for the X-cube model on the minimal structure. Since this $L_x = L_y = L_z = 1$ minimal structure has $\log_2 \text{GSD} = 2$, we can use the foliation structure to claim that

$$\log_2 \text{GSD} = L_x + L_y. \tag{9}$$

One has to be careful while counting logical operators starting from the underlying toric codes. In particular, if we try to count the GSD of the X-cube model with periodic boundaries using the 't Hooft loops of the toric codes, one may naively conclude that all six (two for each plane) loops commute with the effective Hamiltonian and hence are logical operators. But one should note that, because of the $ZZ$ coupling term, all these 't Hooft loops are not independent. In fact, it can be verified that they pair up into equivalent logical operators, which reduces the $\log_2 \text{GSD}$ by 3.

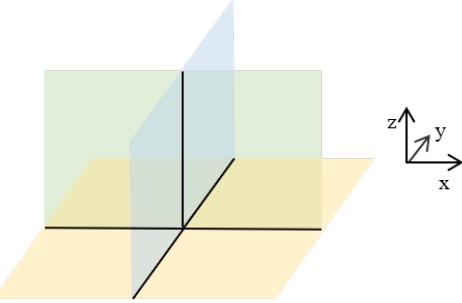

Figure 14: The minimal structure of an X-cube model with rough boundaries.

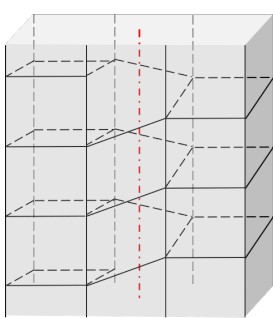

Figure 15: A cross-section of a cubic lattice with a smooth screw dislocation. The dotted red line denotes the defect line, and it cuts through the plaquettes on each plane.

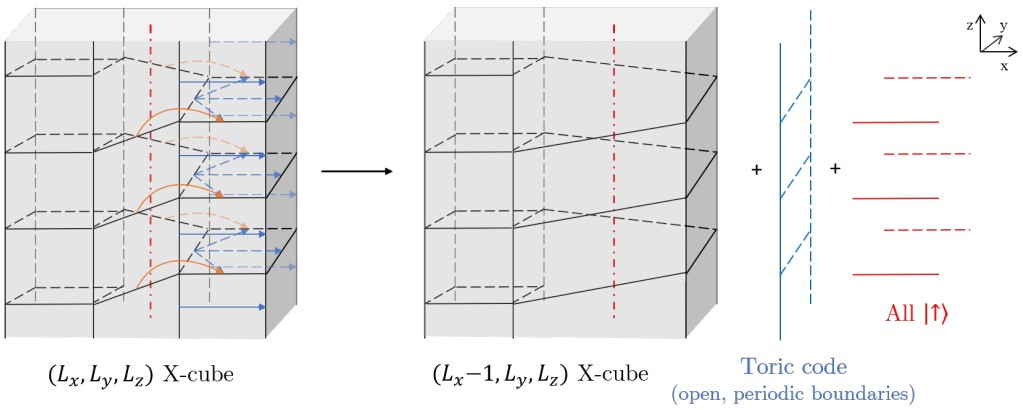

Figure 16: The entanglement RG process to remove a toric code leaf adjacent to a screw dislocation. The arrows denote CNOT gates as in Fig. 10. (Colored for clarity.)

# 3 Smooth Screw Dislocation

In this section, we study the ground state properties of an X-cube model defined on a lattice with a smooth screw dislocation, shown in Fig. 15. We choose the dislocation line to be along the $z$ axis and the lattice has periodic boundary condition along $z$ and open boundaries along $x$ and $y$. The Hilbert space consists of a qubit on each link, and the Hamiltonian is the same as the usual X-cube model except that there are no cube terms along the dislocation line. The vertex terms are well-defined because every vertex on this lattice is locally homeomorphic to a vertex in a simple cubic lattice without the defect. We call this a *smooth* dislocation because it condenses fracton dipoles (which are $xz$- and $yz$-planons), like a smooth boundary. No lineons are condensed at the defect.

## 3.1 Foliation structure

This system has a foliation structure consisting of one $xy$-leaf, which spirals like a circular stairway, and $L_x$ $yz$-leaves and $L_y$ $zx$-leaves, which form perpendicular stacks along their respective directions. We will use the entanglement RG procedure from the previous section to remove the $yz$- and $zx$-leaves to reach the minimal structure of this model. One can see that it is straightforward to remove these leaves away from the screw dislocation—it is the same process as discussed in the previous section. We show in Fig. 16 that the same can be said for leaves adjacent to the dislocation line. Therefore, we can use this RG process to obtain an X-cube model of size $(2, 2, L_z)$ with a screw dislocation, which is the minimal structure.

## 3.2   Increased mobility of quasiparticles

An important consequence of adding a screw dislocation is that a subdimensional excitation in a fracton model can have increased mobility due to the dislocation. First, recall that fracton and lineon dipoles are planons, and that the screw dislocation connects all the $xy$ planes of the original foliation (Fig. 1). This means that we can wind an $xy$ planon around the defect to effectively displace it by one step in the $z$-direction. We call this process the *winding* of a planon around the dislocation. There is an important consequence for this. We note that $xy$ planons have fractons/lineons separated by one step in the $z$-direction. If we look at the operator that winds this around the dislocation once, we see that this operator effectively creates two fractons/lineons separated by *two* steps in the $z$-direction. This means that this winding allows fractons and $x/y$ lineons to hop in the $z$-direction by an even number of steps. We call this process *tunneling* of fractons/lineons along the dislocation line because although some of the intermediate states only involve a single fracton or lineon excitation, these processes involve other intermediate states with higher energy – similar to the tunneling of a quantum particle through an energy barrier. Because of this new mobility, these excitations can now go around the periodic boundaries, which can give us new logical operators for the system.

## 3.3   Smooth outer boundaries

The minimal structure looks like Fig. 17a. The system has no cube terms. The effective Hamiltonian for the ground state include vertex terms of the form $Z_1 Z_2$, $Z_1 Z_3 Z_4$ and $Z_2 Z_3 Z_4$, summed over all vertices (refer Fig. 17a). The ground state degeneracy and the corresponding logical operators can again be figured out from the coupled layer construction. The minimal structure can be thought of as the result of coupling two $xz$, two $yz$ and one $xy$ toric code layers. The $xz$ and $yz$ layers have smooth open boundary in one direction and are periodic in the other ($z$ direction). The $xy$ layer on the other hand spirals around the defect and has smooth boundary condition both at the defect and on the outer boundary. The Wilson loops of the intersecting $xz$ and $yz$ layers combine into the logical operator of the fracton model and they correspond to each of the four vertical blue lines in Fig. 18a. The Wilson loop in the $xy$ layer binds with segments of Wilson lines in the $xz$ and $yz$ layers as it spirals around the defect. The Wilson line segments do not act in the ground space of the $xz$, $yz$ layers but when connected by vertical edges as shown in Fig. 18b, they do. This is of course expected because the Wilson lines when bound together become lineon operators and cannot change direction. If we want it to wind around the defect, there has to be tunneling of $z$ direction lineon, which is realized by the vertical edges in the logical operator of Fig. 18b. Such tunneling is only consistent when

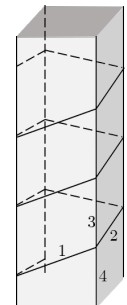

**(a)** Smooth outer boundaries

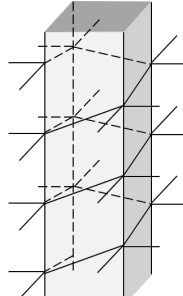

**(b)** Rough outer boundaries

Figure 17: The minimal structure of a smooth screw dislocation.

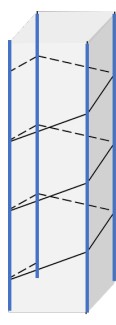

**(a)** Winding logical operator

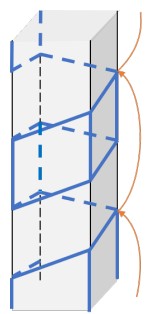

**(b)** Tunneling logical operator (even $L_z$)

Figure 18: The extra logical operators for the X-cube model with a smooth screw dislocation and smooth outer boundaries. The logical operators are products of $X$ over the blue links. The arrow in **(b)** shows the tunneling path of a $x$ lineon that creates the logical operator; intermediate states involve more than one lineon.

$L_z$ is even. The ground state degeneracy of the minimal structure is hence given by

$$\log_2 \text{GSD} = \begin{cases} 4 & \text{for odd } L_z \\ 5 & \text{for even } L_z \end{cases} . \tag{10}$$

This along with the foliation structure tells us that

$$\log_2 \text{GSD} = \begin{cases} L_x + L_y & \text{for odd } L_z \\ L_x + L_y + 1 & \text{for even } L_z \end{cases} . \tag{11}$$

Comparing this to the standard X-cube model (without a dislocation), we see that there are 1 or 2 new logical operators, depending on $L_z$. These logical operators are shown in Fig. 18, and they can be interpreted as moving the lineon dipole around the periodic boundaries by winding, and moving lineons around the periodic boundaries by tunneling (only for even $L_z$) respectively. The winding operator (Fig. 18a) can be seen by referring to Fig. 5b and noting that the operator that winds a lineon dipole around the defect would contain $X$ on all $z$-links, and $X$ applied *twice* (because two lineons have to pass) on every $x$- and $y$-link, which squares to give the identity operator on those links. The tunnelling operator (Fig. 18b) is just the operator that winds a lineon dipole by one step, repeated at every even $L_z$. This gives $X$ applied on every $x$- and $y$-link, and every alternate $z$-link. An interesting subtlety is that this exists only for even $L_z$, as we cannot define "every alternate $z$-link" for odd $L_z$. This is expected, as tunneling hops lineons by two steps, so we need $L_z$ to be even for this process to create a new logical operator. This explains the extra degeneracy we observe for even $L_z$.

The cube logical operators that anticommute with these new logical operators are $xy$-plane rectangular membrane operators with a corner at the screw dislocation and the other corners on the outer boundary. One can verify that, modulo the stabilizer group and the usual cube logical operators (that takes $x$- and $y$- cube dipoles to opposite smooth boundaries), there are one and two such independent operators for odd and even $L_z$, respectively.

## 3.4 Rough outer boundaries

In the case of rough outer boundaries, the minimal structure looks like Fig. 17b. By counting the logical operators, we get

$$\log_2 \text{GSD} = 4 , \tag{12}$$

for the minimal structure. Like in the previous section, one can construct these logical operators starting from the logical operators of the underlying toric codes. Using the foliation

structure, we conclude that

$$\log_2 \text{GSD} = L_x + L_y,\tag{13}$$

for any $L_x, L_y, L_z$. We observe that this has the same GSD as the corresponding X-cube model without the dislocation, and one can verify that the operators that wind planons and tunnel fractons/lineons around the periodic boundaries are all trivial (modulo the stabilizer group). Therefore, there are no extra logical qubits in the ground space.

# 4 Other types of defects

In this section, we consider more general types of dislocation defects and how the ground space of the X-cube model is affected. Our motivation is to obtain a more complete understand of other simple line defects in the X-cube model. We use the prefixes 'rough' and 'smooth' for defects that condense composites of lineon and fracton excitations, respectively.

## 4.1 Edge dislocation

An edge dislocation in the model (see Fig. 20) can be roughly thought of as an X-cube model with a half-plane of toric code added to the foliation structure. When the half plane is added, one boundary of the toric code lies in the bulk of the X-cube model. Depending on the nature of the excitations condensed at the defect line, we classify edge dislocations into smooth and rough.

The procedure to add a smooth edge dislocation to an X-cube model is straightforward: near the defect line one can apply the same local unitary that we used to add a toric code leaf to a system with smooth boundaries in Sec. 2.3.2. This means that to find the $\log_2 \text{GSD}$ of the X-cube model on this lattice, we just have to add the $\log_2 \text{GSD}$ of the toric code half-plane to that of the X-cube model on the standard lattice that we started with.

Adding a rough edge dislocation is slightly more complicated. Naively, one would expect that analogous to the smooth case, a rough edge can be constructed by inserting a half-plane of toric code with a rough boundary; but this is not the case. At the rough edge of a toric code, we know that the vertex excitation, also known as the flux excitation, of the toric code condenses. Under the local circuit used to sew the half-plane into the foliation, the flux excitation transforms into a lineon dipole (which is a planon, so the mobility matches that of a flux in 2D). Therefore, if the X-cube model with a rough dislocation was connected to an X-cube model plus half-plane of toric code via a finite-depth local circuit, the defect must condense

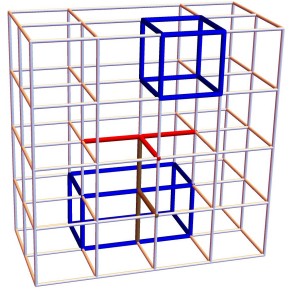

**(a)** Smooth edge dislocation

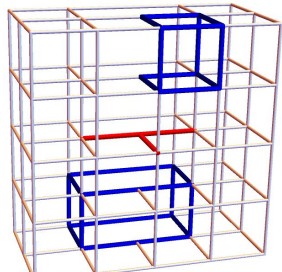

**(b)** Rough edge dislocation

Figure 19: The Hamiltonian terms for the X-cube model with an edge dislocation. $A_c$, $B_v^z$, and $B_v^x$ terms near the defect are highlighted in blue, red and brown respectively. The rough edge dislocation only has one vertex term on vertices on the defect line.

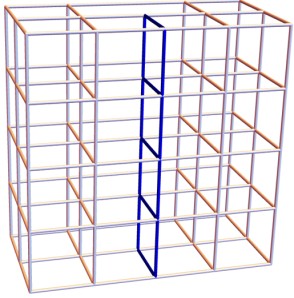

**(a)** Smooth edge dislocation

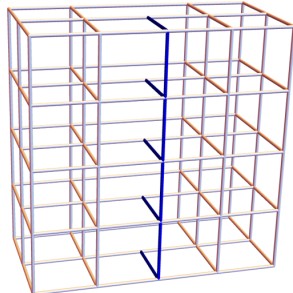

**(b)** Rough edge dislocation

Figure 20: The lattice with edge dislocations as a half-plane added to a simple cubic lattice. The extra half-plane is highlighted.

lineon dipoles. But the system we have defined condenses a pair of $x$ and $y$ lineons at the defect line, which is not a lineon dipole. So to construct this, we start from the smooth edge dislocation and use a (non-local) unitary transformation to remove the qubits on the extra $z$-link as product/free states. It turns out that, effectively, the GSD added to the system because of this process is equal to the GSD of a toric code half-plane with rough boundaries near the defect (and outer boundaries which match the outer boundary of the X-cube model). So, although the rough edge dislocation can not be described using a toric code leaf that is sewn in using a local unitary, the resulting GSD is the same. Therefore

$$\log_2 \text{GSD} = \begin{cases} \log_2 \text{GSD}_\circ + 1 & \text{if the defect and boundary types match} \\ \log_2 \text{GSD}_\circ & \text{if they do not match} \end{cases}, \qquad (14)$$

where the half-plane of toric code is not counted in the system size and $\text{GSD}_\circ$ denotes the GSD of the X-cube model we started with. One can add more half-leaves to get an edge dislocation with a larger Burgers vector, and this can be done by adding toric code half-leaves (with smooth/rough boundary near the dislocation line) using a finite-depth local unitary, and the extra unitary transformation in the case of rough edge dislocations is not required.

## 4.2 Holes

We define a hole as the line-like defect where a line of links (qubits) are missing from the lattice. These defects can be constructed by adding pairs of edge dislocations. The size of a smooth/rough hole is specified by the number of plaquettes or vertices it covers. We have shown the structures of some basic holes in Fig. 21. The $1 \times 1$ smooth hole is trivial; it is just the standard simple cubic lattice.

There can be two kinds of holes respecting translation symmetry along $z$, constructed from adding pairs of smooth and rough edge dislocations. These are smooth and rough holes respectively, because the former can condense fracton dipoles, while the latter can condense composites of lineon excitations. We can imagine having larger holes by adding more edge dislocations.

The construction of edge dislocations by adding half-planes presents us with an easy way to calculate the GSD of a system with holes (Fig. 22). The $\log_2$ GSD of a system with a hole is equal to the sum of the $\log_2$ GSDs of its parts, which are a standard X-cube model and the toric code half-planes. It can be verified that for an $m \times n$ smooth (rough) hole, we need to add $2m + 2n - 4$ half-planes of toric code with appropriate boundaries to a $1 \times 1$ hole. The GSD of some holes have been tabulated in Table 1. Note that in the table, the planes formed by the toric code half-leaves are not counted in the system size. So, in the table, $(L_x, L_y)$ is the system size of the pure X-cube model with boundaries that we added half-planes to. Just like

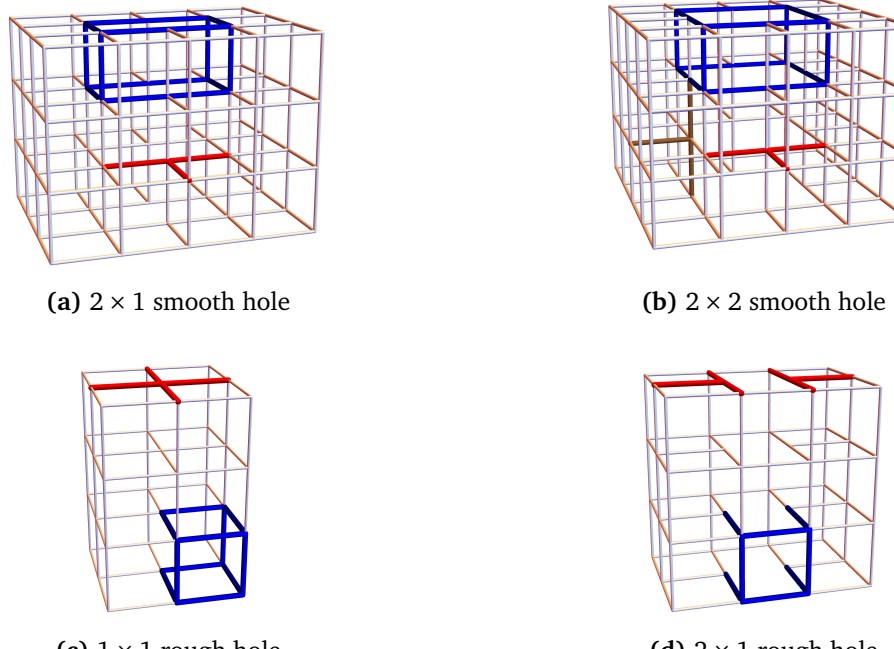

**(a)** 2 × 1 smooth hole

**(b)** 2 × 2 smooth hole

**(c)** 1 × 1 rough hole

**(d)** 2 × 1 rough hole

Figure 21: The X-cube model with holes. The shapes of the nontrivial cube (blue) and vertex (red and brown) terms are highlighted. We note that the Hamiltonian terms at edge of the holes resemble the terms at the smooth and rough outer boundaries. For each type of hole, there is also a cube or vertex term that surrounds the entire hole. Note that in (d), there is just a single disconnected vertex term that is the product of six operators on the six red links.

the case of the edge dislocation, the additional half-leaves added to make larger rough holes need to be sewn in using a finite-depth local unitary.

## 4.3 Rough screw dislocation

A rough screw dislocation is given by a lattice whose dislocation line passes through vertices, which can be constructed by starting with an X-cube model with a 1×1 rough hole and creating a screw dislocation defect about the hole with Burgers vector along $z$. The Hamiltonian is given

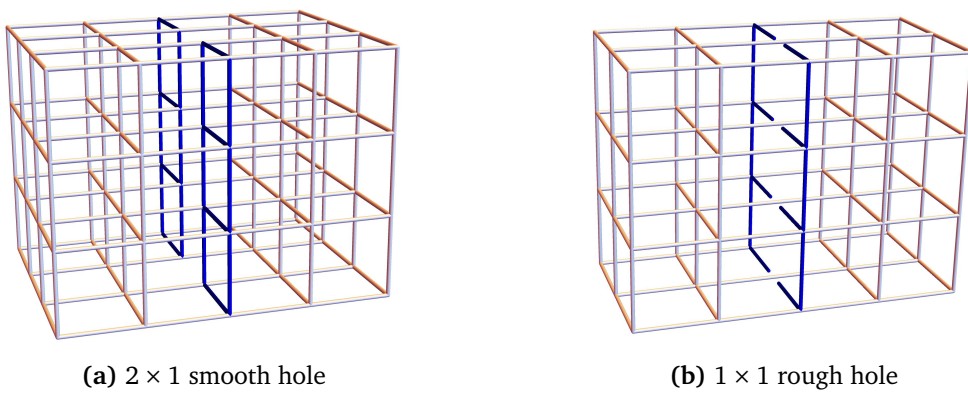

**(a)** 2 × 1 smooth hole

**(b)** 1 × 1 rough hole

Figure 22: Constructing holes by inserting opposing edge-dislocations (toric code half-leaves).

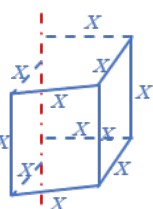

Figure 23: Unlike the smooth screw dislocation, the cube terms in a rough screw dislocation are not all locally homeomorphic to a simple cubic cell. This figure shows the structure of the cube term adjacent to the dislocation (denoted by the red dotted line). All the vertex terms are locally homeomorphic to a vertex in a simple cubic lattice, so we use the natural definition of $B_v^\mu$ in that case. There are no vertex terms on the dislocation line.

by smoothly transforming the Hamiltonian terms for the X-cube model on a lattice with a $1 \times 1$ rough hole, while removing the vertex term surrounding the dislocation line. The way the cube term adjacent to the defect line gets smoothly deformed is shown in Fig. 23. This model condenses $x$ and $y$ lineons at the dislocation line, hence named rough. The foliation structure is similar to the smooth case except that the $xy$-leaf has rough boundaries near the dislocation and the $xz$ and $yz$ leaves passing through the dislocation line are now cut in half. But we can use the same treatment as the previous section to obtain the minimal structure for this model, given in Fig. 24.

Like before, we count the logical operators in the minimal structure. The minimal structure with rough outer boundaries has a single type of cube term $- X_1 X_2 X_3 X_4$ (Fig. 24a) – and the Hamiltonian is the sum of all such cube terms. There are no vertex terms in this minimal structure. We observe that if $L_z$ is even, there are 5 independent pairs of logical operators, whereas there are only 4 for odd $L_z$. Therefore, the GSD of the minimal structure is

$$\log_2 \text{GSD} = \begin{cases} 4 & \text{for odd } L_z \\ 5 & \text{for even } L_z \end{cases} . \tag{15}$$

Therefore, the GSD with a general system size is given by

$$\log_2 \text{GSD} = \begin{cases} L_x + L_y + 2 & \text{for odd } L_z \\ L_x + L_y + 3 & \text{for even } L_z \end{cases} . \tag{16}$$

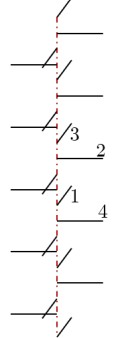

**(a)** Rough outer boundaries

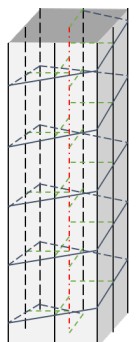

**(b)** Smooth outer boundaries. Links colored for clarity.

Figure 24: The minimal structure of a rough screw dislocation. The dislocation line is colored red.

Comparing this to the GSD of a system with rough boundaries and a $1 \times 1$ hole (without a screw dislocation), which has

$$\log_2 \text{GSD} = L_x + L_y + 1 \,, \tag{17}$$

we have either 1 or 2 extra logical qubits in the ground space. Here, this corresponds to the logical operators that tunnels the fracton (only for even $L_z$) and winds the fracton dipole around the dislocation line and across the periodic boundaries, which arise because of the increased mobility due to the dislocation, which has been discussed in Sec. 3.2.

We can give a similar argument for the case of the smooth outer boundary. In the case, the minimal structure (Fig. 24b) has the cube terms adjacent to the dislocation line (now with 11 links) as well as vertex terms on the smooth outer boundary. The GSD is $2^4$ for the minimal structure. We do not have winding/tunneling logical operators because of the mismatched boundaries. For the entire system, the GSD is

$$\log_2 \text{GSD} = L_x + L_y - 2 \,. \tag{18}$$

## 4.4 Larger screw dislocations

In Sec. 4.2, we looked at holes. The smooth screw dislocation (discussed in Sec. 3) is a simple cubic lattice (equivalent to a system with a $1 \times 1$ smooth hole) with a screw dislocation around the hole. The rough screw dislocation is a cubic lattice with a $1 \times 1$ rough hole containing a screw dislocation. To generalize this idea to larger screw dislocations: we start from the standard X-cube model, create a hole of the appropriate size and type, and add a screw dislocation with Burgers vector along the defect line. When we add the screw dislocation, the Hamiltonian term that surrounds the hole (see Fig. 21) becomes ill-defined and is therefore removed from the Hamiltonian. The rest of the terms get smoothly deformed under the addition of the dislocation, and their sum gives us the Hamiltonian for the X-cube model with a screw dislocation.

The $2 \times 1$ smooth screw dislocation has the lattice shown in Fig. 25. The minimal structure approach could be used to calculate the GSD, but it becomes cumbersome to count the logical operators for the minimal structures of large dislocations. The $2 \times 1$ smooth screw dislocation can be obtained from the $1 \times 1$ smooth screw dislocation by adding a pair of edge dislocations, similar to how we obtained the $2 \times 1$ smooth hole from the X-cube model (which can trivially be thought of as having a $1 \times 1$ smooth hole). Similarly, we find that the X-cube model with a $2 \times 1$ smooth screw dislocation can be obtained from the X-cube model with a $1 \times 1$ smooth screw dislocation and two toric code half-leaves via a local unitary. These toric code half-leaves have

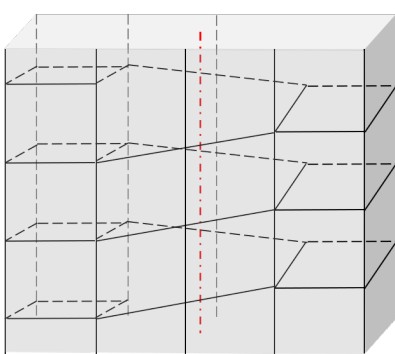

Figure 25: A lattice with a $2 \times 1$ smooth screw dislocation. We see that the dislocation line consists of a row of two adjacent plaquettes, and it is exactly the $2 \times 1$ smooth hole with a screw dislocation.

$$1 \times 1 \text{ Hole} \xrightarrow{\ +1(+1)\ } \begin{matrix} 1 \times 1 \text{ Screw} \\ \text{Dislocation} \end{matrix}$$

$$\downarrow {+2m+2n-4} \qquad\qquad \downarrow {+2m+2n-4}$$

$$m \times n \text{ Hole} \xrightarrow{\ +1(+1)\ } \begin{matrix} m \times n \text{ Screw} \\ \text{Dislocation} \end{matrix}$$

Figure 26: The scheme to create a large screw defect forms a commutative diagram. The horizontal lines add the dislocation about the hole and the vertical lines add half-leaves. The value on the arrows denote the change in $\log_2$ GSD during the process for matching defect and outer boundaries, with the additional (+1) for even $L_z$.

smooth boundaries near the dislocation and matching boundaries near the outer end, similar to Fig. 22a. This tells us that, for the X-cube model with a $2 \times 1$ smooth screw dislocation,

$$\log_2 \text{GSD} = \begin{cases} \log_2 \text{GSD} (\text{X-cube}_{1 \times 1 \text{ dislocation}}) + 2 & \text{if the defect and boundary types match} \\ \log_2 \text{GSD} (\text{X-cube}_{1 \times 1 \text{ dislocation}}) & \text{if they do not match} \end{cases} . \tag{19}$$

We can repeat this process to obtain a screw dislocation of any size. The results for other large screw dislocations have been tabulated in Table 1, and the process is schematically shown in Fig. 26. Note that in the table, the planes formed by the toric code half-leaves are not counted in the system size $(L_x, L_y)$. These results are also consistent with Fig. 2.

## 4.5 Higher-order screw dislocations

We define the order of a screw dislocation to be the magnitude of the Burgers vector in terms of the lattice spacing. All the screw dislocations discussed till now are order-1 dislocations. We can imagine having a screw dislocation of order $n$, which just means if we go around the dislocation once, we end up $n$ steps above (along $z$) where we started from. In the foliation picture, this can be thought of naturally as a stack of $n$ $z$-leaves winding along $z$ like a simple dislocation, which gives us a jump of $n$ for every winding.

If $L_z$ is a multiple of $n$, then we have $n$ independent leaves. We can imagine doing a finite-depth local unitary to take a leaf out, that maps

$$n \longrightarrow n-1 \quad L_z \longrightarrow L_z - \frac{L_z}{n} .$$

The process to exfoliate leaves is similar to what we did in Sec. 2.3.2, where we analysed the X-cube model with open boundaries. Upon doing the RG process to exfoliate one layer, we get a spiralling toric code leaf which is topologically equivalent to a toric code on an annulus (Fig. 1). We know this has the following degeneracy:

$$\log_2 \text{GSD} = \begin{cases} 1 & \text{if the defect and boundary types match} \\ 0 & \text{if they do not match} \end{cases} . \tag{20}$$

By exfoliating layers until we are left with an order $n = 1$ screw dislocation, we can split the GSD of the entire system into a part from the toric code layers (Eq. (20)) and a part from the remaining system with an order $n = 1$ screw dislocation.

If $L_z$ is not a multiple of $n$, then the $m^{\text{th}}$ $xy$-layer is connected to layer number $m + L_z$ (mod $n$) as we go across the periodic boundary in the $z$ direction. Thus, the different layers start to connect with each other and we only have $\text{GCD}(n, L_z)$ independent (disconnected) leaves. We can perform the same exfoliation process to obtain a spiralling toric code, but this time the toric code will have a screw dislocation of order $\frac{n}{\text{GCD}(n,L_z)}$ with the same GSD as in Eq. (20). After exfoliating $\text{GCD}(n, L_z) - 1$ leaves, we are left with an X-cube model of height $L_z' = \frac{L_z}{\text{GCD}(n,L_z)}$ and order $n' = \frac{n}{\text{GCD}(n,L_z)}$.

This is a system we have not encountered before, but the GSD of the minimal structure is the same as that of the minimal structure of the simple screw dislocation, and changes when $L_z'$ is odd/even. This can be verified by going to the minimal structure and counting the logical operators. The logical operators look the same in this case and the simple ($n = 1$) dislocation. Therefore, we find:

$$
\log_2 \text{GSD} =
\begin{cases}
\log_2 \text{GSD}(\text{X-cube with simple dislocation}, L_z') + \text{GCD}(n, L_z) - 1 \\
\qquad\qquad \text{if the defect and boundary types match} \\
\\
\log_2 \text{GSD}(\text{X-cube with simple dislocation}, L_z') \\
\qquad\qquad \text{if they do not match}
\end{cases}
\tag{21}
$$

where $\text{GSD}(\text{X-cube with simple dislocation}, L_z')$ denotes the GSD of the X-cube model with height $L_z'$ and a simple screw dislocation of the same type (smooth or rough) as the the order-$n$ screw dislocation.

We note that, for higher-order screw dislocations, the possibilities for gapped defects go beyond rough and smooth. We expect there to exist additional 'twisted' dislocations in which composites of electric and magnetic excitations condense along the defect line.

# 5 Summary

In this paper, we study the effect of screw dislocations in the X-cube model and show how they reveal nontrivial features in the underlying fracton order. In particular, we find that inserting a screw dislocation can result in a change in ground state degeneracy by a constant factor. Table 1 summarizes the different cases.

The degeneracy change can result from two effects: (1) the winding of planon quasi-particles around the screw dislocation; and (2) the tunneling of fractonic quasi-particles along the screw dislocation (where they gain mobility). These effects change the degeneracy only when the boundary condition at the defect matches with the outside boundary, and the latter effect has an even/odd dependence on the length of the defect line. We expect similar physics to apply more generally to other foliated fracton models.

Although we considered many kinds of line defects, it would be very interesting to obtain a more general and systematic understanding of these defects in fracton models.

It is also interesting to note that if we were to consider a screw defect with a large radius $R \gg 1$, then the logical operator in Fig. 18b would become a surface operator with $O(RL_z)$ support. Therefore, it should have a more robust quantum memory than the logical operators of the X-cube model without screw defects. However, it is conjugate to a string logical operator, which makes this pair of string and membrane logical operators similar (in dimensionality) to those of 3D toric code.

Our calculation makes use of both the foliation structure of the X-cube model and its coupled layer structure. In particular, we decouple 2D toric code foliation layers from the X-cube model until a minimal structure is reached, which can be interpreted as the result of coupling

Table 1: $\log_2$ GSD of different geometries. If there is a '(+1)' suffixed to the result, it means that the given result is for odd $L_z$, while for even $L_z$ there is one more logical qubit because of the spiralling logical operator. If there is a subscript $L_z'$ to the suffix, it means that we should add the +1 to the $\log_2$ GSD if $L_z' = \frac{L_z}{\text{GCD}(L_z,n)}$ is even.

| | Smooth boundaries | Rough boundaries |
|---|---|---|
| No defect (simple smooth hole) | $L_x + L_y - 1$ | $L_x + L_y$ |
| Simple smooth screw dislocation | $L_x + L_y$ (+1) | $L_x + L_y$ |
| Smooth edge defect | $L_x + L_y$ | $L_x + L_y$ |
| $m \times n$ smooth hole | $L_x + L_y - 5$ $+2m + 2n$ | $L_x + L_y$ |
| $m \times n$ smooth screw dislocation | $L_x + L_y - 4$ $+2m + 2n$ (+1) | $L_x + L_y$ |
| Order-$n$ smooth screw dislocation | $L_x + L_y + \text{GCD}(L_z,n)$ $- 1$ $(+1)_{L_z'}$ | $L_x + L_y$ |
| Rough edge defect | $L_x + L_y - 1$ | $L_x + L_y + 1$ |
| Simple rough hole | $L_x + L_y - 1$ | $L_x + L_y + 2$ |
| Simple rough screw dislocation | $L_x + L_y - 1$ | $L_x + L_y + 3$ (+1) |
| $m \times n$ rough hole | $L_x + L_y - 1$ | $L_x + L_y - 2$ $+2m + 2n$ |
| $m \times n$ rough screw dislocation | $L_x + L_y - 1$ | $L_x + L_y - 1$ $+2m + 2n$ (+1) |
| Order-$n$ rough screw dislocation | $L_x + L_y - 1$ | $L_x + L_y + 2$ $+\text{GCD}(L_z,n)$ $(+1)_{L_z'}$ |

a small number of foliation layers together along the intersection line. The GSD of the foliation layers and the minimal structure can each be easily determined. The total GSD is then obtained as their product. More broadly, we know that some type I fracton models have a coupled layer structure [28,29,34,35], and some have a foliation structure. To what extent these two are related to each other is an interesting open problem. Beyond type I models, twisted boundary condition and translation defects can also lead to nontrivial effects [36]. How to interpret these effects is an interesting open question.

Lattice defects, such as edge and screw dislocations, can be described on a Riemannian manifold in terms of (quantized) curvature and torsion. In Ref. [37], it was also shown that fractons in many gapless $U(1)$ fracton models [38–41] can gain mobility in the presence of certain kinds of spatial curvature, somewhat similar to the X-cube fractons near a smooth screw dislocation. Certain kinds of curvature within fractons models has also been studied in Ref. [42,43], where it has also been found that curved lattices can result in X-cube fractons gaining mobility [43] and disclination defects can have connections to hologrpahy [42]. Gapped fracton models (such as the X-cube model) can be defined on a foliated manifold without specifying a metric (for which there is no Riemannian structure) [32,44,45]. In the continuum, a foliation structure can be described in terms of a 1-form foliation field $e_\mu$ [44]. It is plausible that edge dislocations are present wherever $e_\mu$ has nonzero curl (i.e. $\nabla \times e \neq 0$ or $de \neq 0$). However, the study of these effects in the continuum remains as a future direction.

## Acknowledgments

This work initiated from discussion with Alexei Kitaev. We are indebted to inspiring discussions with Jennifer Cano, Arpit Dua, Meng Cheng, and Nathan Seiberg. N.M. acknowledges the KVPY programme. W.S. and X.C. are supported by the National Science Foundation under award number DMR-1654340, the Simons collaboration on "Ultra-Quantum Matter," which is a grant from the Simons Foundation (651440) and the Institute for Quantum Information and Matter at Caltech. K.S. and X. C. are supported by the Walter Burke Institute for Theoretical Physics at Caltech. While working on this project, we learned that Tom Rudelius et. al. [46] and Arpit Dua et. al. are also studying the X-cube model in twisted lattices.

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
