# Peer review of "Screw dislocations in the X-cube fracton model"

_SciPost Physics, doi:SciPost Phys. 10, 094 (2021)_

## Round 2 · Referee Report · Anonymous (Referee 1) · 2021-2-7

Strengths

  1. The authors carefully analyze the X-cube model in the presence of boundaries and/or defects. In particular, they compute the ground state degeneracy and analyze the restricted mobility of the excitations. These are valuable results for further understanding the gapped fracton models.

  2. Their analyses show that the underlying foliation structure, as well as the coupled layer construction, of the X-cube model are not only conceptual advancements but also efficient for practical computations.

  3. The derivations are streamlined and clearly explained. The final results are also concisely summarized.

Weaknesses

  1. Some of the choices of the defects or dislocations are not well-motivated, and it isn't clear why these particular defects are of special importance. Perhaps the authors can motivate these discussions in a more familiar context such as the toric code.

Report

This paper reveals interesting new properties of the X-cube model. It shows that various observables, such as the ground state degeneracy, of the X-cube model, can depend sensitively on the system size in the presence of defects. I would recommend it for publication.

Requested changes

  1. Are there gapped boundary conditions other than the ones discussed in section 2.3? Perhaps the authors can briefly summarize the status of classifying gapped boundary conditions for the X-cube model along the line of [31]. This addition would motivate the consideration of the smooth and rough boundary conditions better.

  2. Can some of these defects also be studied with periodic boundary conditions in all directions?

  3. Above (9), " t' Hooft loop " -> " 't Hooft loop".

  4. In the paragraph above section 3.4, the font of "xy-plane" is not consistent with the rest of the paper.

  • validity: high
  • significance: high
  • originality: high
  • clarity: high
  • formatting: excellent
  • grammar: excellent

Author:  Kevin Slagle  on 2021-04-16  [id 1365]

(in reply to Report 1 on 2021-02-07)

Weaknesses: Our main focus was on screw dislocations (studied in Sec 3), which we motivate in the second paragraph of the introduction. The defects in Sec 4 were chosen since they are some of the simplest examples of other line defects in the X-cube model. In a 3D toric code, none of these defects affect the long distance physics. However, it is useful to think about these defects within the X-cube model as defects in the foliation, which does result in modifying the foliating 2D toric code layers. We appeal to this intuition many times throughout Sections 3 and 4.

Requested changes: 1) Yes, indeed Ref. [31] discusses four different X-cube boundary conditions. The first two are the smooth and rough boundaries that we discuss. The other two can be interpreted as mixed smooth and rough boundaries. We just chose the two simplest kinds of X-cube boundary conditions. We think that choosing the other two boundary types would complicate the discussion without adding too much extra insight. We now clarify this in Sec. 2.3.1.

2) For screw and edge dislocations to have periodic boundary conditions in all directions, one must have a ``dipole'' of screw/edge dislocations such that the total defect has zero Burgers vector. This is a more complicated object which we have not discussed in this manuscript, but it could be an interesting system for future study. Only the hole defect could be studied with periodic boundaries in all directions.

3,4) Thank you for carefully reading our paper. We have fixed these mistakes.

---

## Round 2 · Referee Report · Anonymous (Referee 2) · 2021-3-1

Strengths

1. The work explores an interesting and important, yet mostly unexplored, aspect of fracton topological orders, namely their translation symmetry lattice defects.

2. The paper is clearly explained and illustrated through many figures.

3. It demonstrates a useful application of the foliation/exfoliation structure of the X-cube model.

Weaknesses

1. The authors seem to only study a restricted set of line defects and there doesn’t appear to be a systematic procedure to address/classify fully general line defects in the X-cube (although I am sympathetic that this may be an intractable problem).

Report

I found this work to be a clearly explained exploration of several types of 1D defects along a lattice direction in the X-cube model that were previously unstudied. The explanation is aided by many nicely illustrated figures. The main tool relied on for analysis of the defect examples is a decoupling or exfoliation procedure previously identified for the X-cube model that allows it to be decomposed into decoupled toric code layers and a minimal leftover structure.
The work is a nice addition to the fracton literature, which has attracted a fair amount of attention in recent years, and I recommend publication by SciPost.

Requested changes

I have a number of suggestions that the author's may consider commenting on:

1. In section 2.2 where the minimal X-cube after exfoliation down to three layers of coupled toric codes is explained, is it possible to represent the coupling of these 3 layers together via a topological defect/domain wall where they meet?

2. In section 3.2, why choose the name tunneling for the mobility of fractons/lineons via a hopping operator adjacent to the defect? (It doesn’t seem like what we would usually call quantum tunneling through a potential barrier.)

3. What about more general line defects that turn corners etc?

4. In a previous work Bulmash-Iadecola found the direction of X-cube boundaries in the lattice could lead to very different properties, is this also true of line defects? I.e. if they are not along a lattice direction would that lead to drastically different characteristics?

5. Do the authors have any comment on the potential utility of the introduced defects for topological quantum memory/computation?

6. Related to the above, in toric code defects and boundaries are often introduced via measuring out qubits in a local basis, can we induce the X-cube defects in this way?

7. Do the authors know if their defects exhaust all possibilities (up to some LU equivalence) for line defects that may be introduced (with the same fixed burgers vector ) via removing all Hamiltonian terms in a neighborhood of the defect and then adding back in a maximal set of local commuting terms to gap the defect?

8. Can such defects be described/classified via the topological defect network construction that was recently introduced by a team including one of the authors on this work?

9. Typo? “six 6 operators on the 6 red links”

  • validity: high
  • significance: good
  • originality: good
  • clarity: high
  • formatting: good
  • grammar: excellent

Author:  Kevin Slagle  on 2021-04-16  [id 1366]

(in reply to Report 2 on 2021-03-01)

Weaknesses: Indeed, we agree that it would be useful to have a more systematic procedure to understand general line defects in the X-cube model. We now note this in the conclusion and leave this problem to future work.

Requested changes: 1) Yes, it is possible in the natural way. Thank you for pointing this out as it seems that we forgot to mention this fact. We added a footnote.

2) We modified the paper to explain this terminology more clearly. In Fig 18b, we show the tunneling operator, which tunnels a single lineon along the vertices marked by the orange arrow. These vertices are special, because moving the lineon between these points will involve intermediate states with larger energy (i.e. more than a single lineon), and in this sense, we call it "tunneling". Thank you for pointing this out.

3) This is an interesting question that we did not consider. Perhaps future work could address this.

4) Indeed, we only considered line defects in the x, y, or z directions. We expect that line defects in diagonal directions would have drastically different characteristics since these directions would not be commensurate with the subdimensional excitations. This would also be an interesting direction for future work.

5) This is a very important question that we thought about several times. One idea was to consider the logical operator in Fig 18b, but where the screw defect has a large $O(L)$ radius where $L$ is the system length. The support of this operator would then scale like $O(L^2)$, which is larger than the $O(L)$ support of the string operators. However, there is a string operator with $O(L)$ support that is conjugate to this surface $O(L^2)$ operator. Thank you for bringing this up; we now mention this potential application in the conclusion.

6) That is an interesting approach. Indeed, it is reasonable that this could be possible for holes and edge dislocations, but it does not seem to be the case for screw dislocations.

7) Our defects do not exhaust all possibilities of line dislocations (up to LU equivalence). For example, the hole defects are just a hole with smooth or rough boundaries. However, one could also image a hold with (em) or (me) boundaries, in the notation of Ref. [33]. More generally, we expect that there are many kinds of defects with increasingly large unit cells.

8) This is an interesting question. We believe that all of the defects we considered can be described by topological defect networks, but we have not checked carefully. This could be an interesting direction for future work, which may be useful for obtaining a more systematic description of these defects.

9) Thank you for the careful reading. We have fixed the typo.

---

## Round 3 · Author Response

We would like to thank the editor and referees for reviewing our manuscript. We are grateful of the positive reviews, useful feedback, and interesting questions. We have replied to the referee reports using SciPost comments and have improved our manuscript with their suggestions.

---

## Round 3 · List of Changes

Clarified motivation of various kinds of line defects.
Added footnotes 1 and 2 to further elaborate on connections to other works.
Clarified our use of tunneling terminology
Added paragraph to conclusion about quantum memory.
Fixed typos.
A detailed diff of the changes can be found here:
https://drive.google.com/file/d/1PFjDXulBwtPD91MOw-UoGt5NZ8ujGY6i

---

## Editorial Decision

published